# Bariatric Surgery Outcomes in Patients with Kidney Transplantation

**DOI:** 10.3390/jcm11206030

**Published:** 2022-10-13

**Authors:** Adriana Pané, Alicia Molina-Andujar, Romina Olbeyra, Bárbara Romano-Andrioni, Laura Boswell, Enrique Montagud-Marrahi, Amanda Jiménez, Ainitze Ibarzabal, Judith Viaplana, Pedro Ventura-Aguiar, Antonio J. Amor, Josep Vidal, Lilliam Flores, Ana de Hollanda

**Affiliations:** 1Endocrinology and Nutrition Department, Hospital Clínic de Barcelona, 08036 Barcelona, Spain; 2Centro de Investigación Biomédica en Red de la Fisiopatología de la Obesidad y Nutrición (CIBEROBN), Instituto de Salud Carlos III (ISCIII), 28029 Madrid, Spain; 3Fundació Clínic per a la Recerca Biomèdica (FCRB), 08036 Barcelona, Spain; 4Nephrology and Kidney Transplantation Department, Hospital Clinic, 08036 Barcelona, Spain; 5Laboratori Experimental de Nefrologia i Trasplantament (LENIT), CRB CELLEX, Fundació Clinic, IDIBAPS, 08036 Barcelona, Spain; 6Endocrinology and Nutrition Department, Althaia, Universitary Health Network, 08243 Manresa, Spain; 7Institut d’Investigacions Biomèdiques August Pi Sunyer (IDIBAPS), 08036 Barcelona, Spain; 8Gastrointestinal Surgery Department, Obesity Unit, Hospital Clínic de Barcelona, 08036 Barcelona, Spain; 9Centro de Investigación Biomédica en Red de Diabetes y Enfermedades Metabólicas Asociadas (CIBERDEM), 28029 Barcelona, Spain

**Keywords:** bariatric surgery, kidney transplantation, obesity, renal graft dysfunction

## Abstract

Obesity and kidney transplantation (KTx) are closely related. Obesity increases the risk of chronic kidney disease and can be a relative contraindication for KTx. Besides, KTx recipients are predisposed to obesity and its comorbidities. Consequently, bariatric surgery (BS) emerges as a powerful therapeutic tool either before or after KTx. Since evidence regarding the best approach is still scarce, we aimed to describe renal and metabolic outcomes in a single centre with more than 15-year experience in both surgeries. Methods: A retrospective study including patients who had received a KTx either before or after BS. Usual metabolic and renal outcomes, but also new variables (as renal graft dysfunction) were collected for a minimum follow-up of 1-year post-BS. Results: A total of 11 patients were included: *n* = 6 (BS-post-KTx) and *n* = 5 (BS-pre-KTx). One patient was assessed in both groups. No differences in the main outcomes were identified, but BS-post-KTx group tended to gain more weight during the follow-up. The incidence of renal graft dysfunction was comparable (4/6 for BS-post-KTx, 3/5 for BS-pre-KTx) between groups. Conclusions: BS in patients with KTx appears to be safe and effective attending to metabolic and renal outcomes. These results seem irrespective of the time course, except for weight regain, which appears to be a common pattern in the BS-post-KTx group.

## 1. Introduction

Obesity has strongly been related to renal function decline regardless of other obesity-associated diseases [1]. Actually, obesity can lead to end stage renal disease (ESRD) and in this context, kidney transplantation (KTx) represents the best renal replacement therapy [2]. Furthermore, obesity can be a relative contraindication for KTx and patients following transplantation are at increased risk of obesity and its comorbidities, as well as new-onset diabetes mellitus after transplantation (NODAT) [3].

Bariatric surgery (BS) is considered the most effective therapeutic strategy in order to achieve and maintain weight loss in severe obesity. This assumption can be challenging in the setting of KTx. Although there is not a general agreement on the body mass index (BMI) cut-off point to discourage KTx, most guidelines recommend weight loss prior to KTx [4]. Therefore, BS may be used as a bridge to KTx and could improve posttransplant metabolic consequences mainly due to immunosuppression [5,6]. When considering BS in kidney recipients, other aspects should be considered. Firstly, a higher risk of surgical complications could be expected and secondly, the absorption of immunosuppressive therapy may be affected [7,8]. Despite these concerns, BS following KTx could also prevent the negative impact of obesity comorbidities on patient’s health status [9].

At the present moment, BS in the KTx setting is gaining much attention. Actually, a clinical practical guideline by the DESCARTES Working Group of ERA is just available [10]. However, there are still multiple unanswered questions. A recent meta-analysis has stated that BS may be safe and effective either prior or following KTx. Nonetheless, the included studies were heterogeneous and thus, the power to draw firm conclusions was limited [11]. Another recently published clinical review points out the benefits of BS in KTx, but without any clear consensus regarding its optimal timing [12]. Little is known about how BS impacts the bioequivalence of tacrolimus currently available commercial products and immunosuppression pharmacokinetics.

Considering this background, we aimed to describe both usual and new renal and metabolic outcomes; for instance, a glomerular filtration rate below 25 mL/min/1.73 m^2^ was used as a surrogate marker for renal graft dysfunction [13], in patients who received a KTx either before or after BS in a single centre with long-experience in both surgeries. Additionally, surgical complications of both procedures and immunosuppression therapy fluctuations were also recorded.

## 2. Materials and Methods

We retrospectively evaluated all patients followed at the Obesity Unit of our institution and eligible for BS, who had already received a KTx or where in list for it, between January 2005 to December 2017. The indications to undergo BS were based on the Clinical practice guidelines of the European Association for Endoscopic Surgery (EAES) on bariatric surgery published at that moment [14]. Although there are no absolute contraindications for BS, relative contraindications include severe heart failure, unstable coronary artery disease, end-stage lung disease, active cancer, drug/alcohol dependency, impaired intellectual capacity, and severe and untreated mental disorders. Non-specific contraindications were applied when considering those patients on waiting list for KTx or who had already received a KTx. The number of BS during the study period were 2296 (mean age 45.15 ± 9.84 years, mean BMI 45.82 ± 5.96 Kg/m^2^): 19 patients (0.82%) met criteria for CKD stages 3 to 4, and 11 patients (0.48%) were in list for KTx or had already received a KTx.

Although the focus of our research were those patients eligible for weight-loss surgery followed at the Obesity Unit, we also revised the number of KTx during the same study period. A total of 2223 KTx were performed: weight and height records were available for 88.0% individuals (*n* = 1955). The mean age was 51.80 ± 11.96 years and the mean BMI was 24.47 ± 4.16 Kg/m^2^. Of the 1955 patients: 92 (4.71%) were within the underweight range (BMI < 18.5 Kg/m^2^), 1079 (55.19%) had a normal weight, 605 (30.95%) overweight (BMI 25.0–29.9 Kg/m^2^), and 179 (9.16%) were in the obesity category (BMI ≥ 30 Kg/m^2^) − 140 (7.16%) were classified as class I obesity (BMI 30.0–34.9 Kg/m^2^), 31 (1.59%) as class II (BMI 35.0–39.9 Kg/m^2^), and 8 (0.41%) as class III (BMI ≥ 40 kg/m^2^)-. Most patients who fulfilled BS criteria before KTx were still eligible for BS 1-year after KTx: 52 (2.86%) individuals had class II obesity and 10 (0.55%), had a BMI ≥ 40 kg/m^2^. A total of 42 patients were appropriate candidates for BS, but only 1/3 were referred to the Obesity Unit (details can be found in Appendix A).

Eleven patients were included and subclassified in Bariatric Surgery after Kidney Transplantation (BS-post-KTx) (*n* = 6), and Bariatric Surgery before Kidney Transplantation (BS-pre-KTx) (*n* = 5). One patient was assessed in both groups, since he received a KTx prior to BS and afterwards required another one due to kidney graft loss. One patient was never included in the BS-post-KTx group since he abandoned the follow-up in our centre. A patient who had a gastric band before KTx was not considered for our analysis. The follow-up was extended to 5-years post-BS for the BS-post-KTx group, and 2-years post-KTx for BS-pre-KTx group.

Because of the reduced group sizes, differences on main outcomes were explored using non-parametric tests (STATA/IC 15.0 [StataCorp.; College Station, TX, USA]) and a descriptive approach was prioritized.

## 3. Results

### 3.1. Subjects’ Characteristics

Characteristics of study participants are detailed in Table 1. All bariatric procedures were sleeve gastrectomy (SG), but one patient in the BS-post-KTx group required Roux-en-Y gastric bypass (RYGB) because of severe gastroesophageal reflux.

Regarding ESRD etiology, the most common causes in the BS-post-KTX group (*n* = 6) were glomerulonephritis (2/6), followed by reflux nephropathy (1/6), hypertension (1/6), renal polycystosis (1/6), and Laurence-Moon-Bardet-Biedl syndrome (1/6); in the BS-pre-KTX group (*n* = 5), the main causes were diabetic nephropathy (2/5), glomerulonephritis (1/5), reflux nephropathy (1/5), and renal polycystosis (1/5). Renal biopsy was performed only in those patients classified as glomerulonephritis (*n* = 4): 75% had IgA nephropathy, and 25%, mesangial proliferative glomerulonephritis. Regarding kidney donors’ origin, 5/6 in the BS-post-KTx group and 4/5 in the BS-pre-KTX group corresponded to deceased donors; the remaining corresponded to living-donors (haploidentical parent-to-child transplantations). As stated before, one patient affected with reflux nephropathy was included in both groups: in both occasions, the renal graft was from a deceased donor.

### 3.2. Metabolic Outcomes following Bariatric Surgery and Kidney Transplantation

The median time lapse between KTx and BS was 4.7 (3.7–5.7) years for BS-post-KTx group, and 1.1 (0.9–1.2) years for the BS-pre-KTx. All the details concerning metabolic outcomes are presented in Appendix A.

In the BS-post-KTx group, obesity comorbidities and laboratory tests results, clearly worsened following KTx. Actually, at the moment of BS, their body weight had increased up to 27%. BS allowed significant weight loss during the first year (110.7 to 79.3 Kg, *p* = 0.028), but weight regain was a common concern at the 5-year follow-up (79.3 to 88.5 Kg, *p* = 0.249). Regarding comorbidities, no significant clinical or analytical deterioration was observed (Appendix A).

Moving onto the BS-pre-KTx group, at 2-years after KTx, body weight remained unchanged. However, the prevalence of hypertension increased from 40% to 80%. Glucose and HbA1c levels also tended to increase, though diabetes incidence at this time-point did not change (Appendix A).

When considering metabolic outcomes in both groups, the most certainly outstanding result corresponds to the BMI trajectory. Although the weight loss achieved following BS was comparable, weight gain was the rule in the BS-post-KTx. Conversely, patients in the BS-pre-KTx group tended to maintain their weight after KTx. Figure 1 shows BMI trajectory depending on the time course after each intervention: BS or KTx. As depicted, when the first surgery corresponds to KTx, there is a clear weight increase following this surgical procedure (BS-post-KTx; solid-line). On the contrary, when the first surgery corresponds to BS, body weight decreases and no weight gain pattern is observed post-KTx (BS-pre-KTx; dashed-line).

### 3.3. Renal Outcomes following Bariatric Surgery and Kidney Transplantation

Kidney allograft rejection following BS occurred in 2 patients in the BS-post-KTx group (a median of 38.7 months after BS), and 1 patient in the BS-pre-KTx group (18.6 months after BS). When kidney allograft dysfunction was detected, it always occurred after BS (except for one case in the BS-post-KTx group, 6.8 months before BS). In this case, renal biopsy informed obesity-related glomerulopathy.

Further details are presented in Table 2.

### 3.4. Immunosuppression after Kidney Transplantation

When analysing the immunosuppressive therapy, one curiosity emerged, as the type of tacrolimus (FK) formulation changed depending on the group. Patients in the BS-post-KTx received Prograf^®^ (an immediate-release oral formulation [Astellas Ireland Co., Ltd., Killorglin, Ireland]) in 2/3 cases. On the contrary, patients in the BS-pre-KTx received Advagraf^®^ (an extended-release oral formulation [Astellas Ireland Co., Ltd., Killorglin, Ireland]) more often than Prograf^®^: 4/5 vs. 1/5.

The dose (mg/day) and levels (ng/mL) of calcineurin inhibitors registered in the 6 months before and after BS for the BS-post-KTx group remained relatively stable for FK: a dose of 2.2 (1.5–11.5) vs. 2.5 (1.3–7.0) mg/day; and levels of 7.4 (4.7–10.0) vs. 5.3 (4.8–7.4) ng/mL, pre- and post-BS, respectively. However, cyclosporin A (CysA) levels changed: a dose of 100 (50–125) vs. 100 (50–142) mg/day; and levels of 64 (60–105) ng/mL vs. 138 (101–147) ng/mL, pre- and post-BS respectively.

## 4. Discussion

In the last decades, many studies, systematic reviews and metanalyses have explored the role of BS on KTx [11,12]. Furthermore, it has to be highlighted that data regarding simultaneous robotic KTx and BS for patients with severe obesity and ESRD has been recently published [15]. Therefore, KTx and BS is a rapidly expanding area of research. However, drawing certain conclusions results challenging because of the completely different approaches which have been used. In fact, even the most recent clinical practice guideline attending the management of obesity in KTx candidates and recipients by the DESCARTES group has identified multiple areas of uncertainty [10].

The vast majority of publications to date have evaluated BS as a bridge to KTx in patients with ESRD [5,16,17,18]. In spite of the heterogenous populations included in each study, they all share a firm statement: BS can reduce the prevalence of obesity comorbidities, and thereby, optimize patients before KTx. In fact, our results strengthen this assumption.

Other groups have assessed the impact of recipient obesity on outcomes after KTx [19]. Nevertheless, hardly any of them have focused on the metabolic outcomes after KTx in those patients with previous BS [17,20,21]. When considering the BS-pre-KTx group, no changes in glucose or lipid profile were observed in our study sample. Besides, the weight loss achieved after BS could be maintained during a median follow-up of 5 years after KTx. In line with previous authors, these results reinforce the power of BS prior to KTx in order to avoid weight regain and obesity comorbidities incidence following solid organ transplantation, along with NODAT. Attending BS-post-KTx, our data supports this option as a validate way in order to improve patient’s metabolic status following KTx in the short to medium term.

Moving onto the renal field, some studies have put their efforts in determining BS safety and efficacy after KTx [22,23,24]. Although the different criteria used in order to define renal outcomes, each study group provides data supporting BS after KTx. Altered immunosuppression absorption and subsequent allograft rejection/dysfunction are common theorical concerns for BS in KTx recipients. However, no consistent data has been reported so far. In our series, the percentage of post-operatory complications, allograft rejection and dysfunction were similar in both groups. No clinically significant changes were observed regarding the levels of FK in the BS-post-KTx group. As stated by Cohen and colleagues [20], this may in part be because individuals who had BS before KTx in our cohort had almost always had SG. Conversely, the levels of CysA showed an irregular pattern. The different calcineurin inhibitors’ pharmacokinetics may explain this observation.

Another aspect that should be mentioned is the relatively low percentage of obesity in our cohort (9.16%), which differs from the prevalence stated by other European groups as for example 15.90% in France [25]. There are several reasons that might explain this discrepancy. First of all, as stated by other study groups, obesity could be an obstacle to transplantation access. Actually, the prevalence of obesity in the dialysis patients of our area is 14.30%. However, it has to be considered that the sociodemographic features of the patients in dialysis are not equal to those enlisted for KTx, being the first ones older (mean age 66.3 ± 14.3 years) [26]. Secondly, the specific characteristics of our Renal Unit, in which highly specialized dietitians offers dietary advice both before and after KTx, might have also contributed to this low percentage. Although an American study showed that among patients with obesity who were required to achieve a BMI < 30 kg/m^2^ before KTx, only 5% reached the final target with a non-surgical approach [27], in our centre, nutritional advice is offered to all the patients who are in special need of weight control, not only those with obesity, but also those who are overweight. Therefore, offering nutritional advice to the whole group of patients with KTx may contribute to preventing obesity, especially severe obesity. Finally, it should also be highlighted that only 1/3 of patients eligible for BS were referred to the Obesity Unit. As non-specific guidelines for the management of obesity in KTx candidates and recipients had been published during the study period, it is possible that few healthcare professionals were well-awared of BS as a valid and safe option to treat obesity in the KTx group.

Our study has strengths and limitations. Firstly, our research not only emphasizes on renal results following BS, but also metabolic outcomes either after BS and/or KTx. Additionally, novel endpoints such as allograft dysfunction were introduced. Lastly, the inclusion of KTx recipients both before and after BS in a single centre with more than 15-year experience in both surgeries may provide better reproducible results for each surgical time-course.

However, limitations should also be acknowledged. The main drawback of our study is its small sample. Consequently, extrapolating our results to other populations has to be taken with great caution, and the results obtained through statistical tests just offer an orientation.

To summarize, BS before KTx not only is a known powerful tool to facilitate active transplant listing, but could also prevent the common weight gain pattern after KTx. Attending BS-post-KTx, although concerns regarding its security may emerge, the current data do not discourage this option. Therefore, the professional community should consider BS for kidney transplant recipients as a valid strategy to achieve weight-loss and obesity comorbidities resolution.

## Figures and Tables

**Figure 1 jcm-11-06030-f001:**
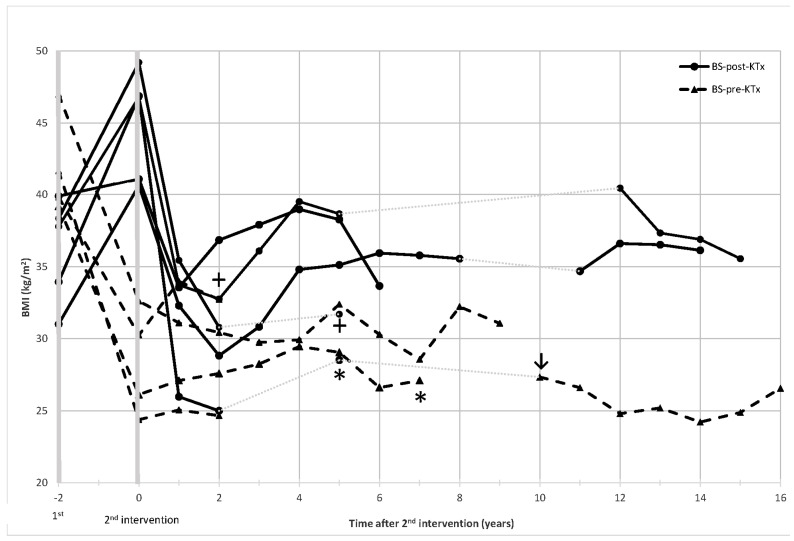
Body mass index trajectories after bariatric surgery (BS) or kidney transplantation (KTx). The 1st surgery refers to KTx for BS-post-KTx, and to BS for BS-pre-KTx. The 2nd intervention refers to BS for BS-post-KTx group, and to KTx for BS-pre-KTx. The time lapse (from 1 to 5 years) between both interventions is not scaled. The cross (+) indicates death and the asterisk (*), renal replacement therapy. The arrow (↓) indicates that one patient categorized into BS-post-KTx group received a 2nd KTx following BS (reclassification into BS-pre-KTx group).

**Table 1 jcm-11-06030-t001:** General clinical characteristics before and 1 year-after bariatric surgery for both groups.

BS-Post-KTx *n* = 6	Baseline	1 Year after BS	*p* Value
Sex (female) *n* (%)	3 (50.0)		-
Age (years)	48.1 (28.6–53.1)		-
Weight (Kg)	110.7 (104.0–120.0)	79.3 (66.5–94.7)	0.027
BMI (Kg/m^2^)	43.9 (40.6–46.9)	32.93 (26.7–33.7)	0.028
TWL (%)	-	26.3 (20.5–28.0)	-
Hypertension, *n* (%)	5 (83.3)	5 (83.3)	0.999
Diabetes, *n* (%)	1 (16.7)	1 (16.7)	0.999
Serum creatinine (mg/dL)	1.40 (1.1–1.7)	1.53 (1.1–1.6)	0.500
eGFR (mL/min/1.73 m^2^)	45.1 (40.5–62.9)	47.8 (45.6–58.6)	0.500
Glucose (mg/dL)	94 (86–109)	84 (77–90)	0.459
Haemoglobin A1c (%)	5.4 (5.15–5.6)	4.8 (4.8–5.1)	0.090
Total Cholesterol (mg/dL)	195 (180–232)	203 (169–210)	0.917
LDL-Cholesterol (mg/dL)	86 (79–143)	113 (103–116)	0.686
Triglycerides (mg/dL)	205 (158–309)	103 (84–118)	0.046
**BS-pre-KTx *n* = 5**	**Baseline**	**1 year after BS**	***p*** **value**
Sex (female), *n* (%)	1 (20.0)		-
Age (years)	38.6 (34.7–61.1)		-
Weight (Kg)	129.0 (120.0–135.0)	81.0 (80.5–82.0)	0.043
BMI (Kg/m^2^)	41.7 (39.8–46.9)	26.8 (25.9–27.4)	0.043
TWL (%)	-	40.4 (31.7–41.7)	-
Hypertension, *n* (%)	4 (80.0)	1 (20.0)	0.250
Diabetes, *n* (%)	3 (60.0)	2 (40.0)	0.999
Glucose (mg/dL)	92 (77–97)	86 (72–88)	0.345
Haemoglobin A1c (%)	5.9 (5.8–7.2)	5.5 (4.8–5.7)	0.080
Total Cholesterol (mg/dL)	182 (133–208)	194 (183–247)	0.500
LDL-Cholesterol (mg/dL)	79 (63–161.5)	120 (105.5–151)	0.715
Triglycerides (mg/dL)	163 (122–356)	132 (118–182)	0.225

Data are shown as *n* (%), median (Q1–Q3). TWL: total weight loss; BMI: body mass index; eGFR: estimated glomerular filtration rate according to CKD-EPI equation. *p* value < 0.05 for group comparisons. Renal function data are not shown in the BS-pre-KTx group because all the patients were on hemodialysis.

**Table 2 jcm-11-06030-t002:** Surgical procedure’s characteristics, complications and follow-up for BS-post-KTx and BS-pre-KTx groups.

	BS-Post-KTx *n* = 6	BS-Pre-KTx *n* = 5
**Bariatric surgery parameters**			
Length of stay (days)	4.5 (3.9–5.9)	6.5 (3.5–8.8)
Surgical time (min)	90 (85–90)	110 (85–130)
**Post-operative complications and clinical intercurrences**		
Early complications (<30 days)	1 (16.7)	1 (20.0)
Surgical site infection	1 (16.7)	0 (0)
Nausea and vomiting requiring IV fluids	0 (0)	1 (16.7)
Late complications (>30 days)	0 (0)	0 (0)
**Kidney transplantation parameters**		
Length of stay (days)	12.0 (8.3–15.2)	16.7 (9.7–20.3)
**Post-operative complications and clinical intercurrences**		
Acute tubular necrosis	0 (0)	1 (20.0)
Delayed graft function	1 (16.7)	0 (0)
**Renal graft function and survival**		
Rejection before BS	2 (33.3)	-
Time since transplantation (months)	1.9 (0.5–3.3)	-
Rejection after BS	2 (33.3)	1 (20.0)
Time since transplantation (months)	71.9 (65.4–78.5)	7.1
Time since bariatric surgery (months)	38.7 (33.9–43.4)	18.6
Graft Dysfunction **#**	4 (66.7)	3 (60.0)
Need for renal replacement therapy	2 (33.3)	1 (20.0)
Time since transplantation (years)	13.4 (8.00–17.1)	7.1 (0.7–10.6)
Time since bariatric surgery (years)	6.1 (3.6–14.2)	8.3 (1.6–11.2)
**Mortality**		
*n*/cause	1 (16.7) COVID-19	1 (20.0) septic shock ^¢^
Time since bariatric surgery (years)	6.2	3.1
Time since renal transplant (years)	10.6	2.1

Data are shown as *n* (%). # eGFR < 25 mL/min/1.73 m^2^ was used in order to define renal graft dysfunction. ^¢^ Septic shock was neither related to BS or KTx.

## Data Availability

The data that support the findings of this study are available from the corresponding authors, A.d.H. and A.P., upon reasonable request.

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
