# Peer review of "Bariatric Surgery Outcomes in Patients with Kidney Transplantation"

_jcm, 2022, doi:10.3390/jcm11206030_

Round 1
Reviewer 1 Report
the paper is well written, easy-to-read and conclusions consistent with Authors' assumptions, as current literature on the same topic is "unbalanced" towards single approach (BS per KTx e.g.). Authors reported a really comprehensive and truly interdisciplinary experience in both fields (ktx and BS), that's the real added value of this paper. The seemingly better effect of BS in pre - transplant setting needs further confirmation.
Intereting paper regarding the "interplay" between BS and KTx.
No significant comment but I would like to point out the limited scientific yield coming from the small sample size, as mentioned by Authors.
Author Response
We are so grateful for your comments. We certainly agree with your point regarding the small sample size, which is a handicap. However, as many areas of uncertainty concerning the management of obesity in kidney transplant candidates and recipients have been identified by the main National and International Societies of Nephrology, we truly believe that our study may contribute to the scientific community.
Reviewer 2 Report
Pané et al in their study titled ‘Bariatric surgery outcomes in patients with kidney transplantation’ carried out a retrospective study to compare metabolic profiles of patients who had received a kidney transplantation (KTx) either before or after BS. The idea notwithstanding the authors fail to impress the importance of carrying out this study and there are major concerns.
The introduction can be improved by adding points for the discussion that can serve as a background. The authors need to explain their results better and highlight the importance of their findings. The write up from 107 to 115 is confusing. Figure 1, Table 1 and Table 2 could be represented in a better manner. In the present form they are very confusing The eGFR and Sr Creat has been mentioned for the post KTx group and not in the pre KTx group.
The discussion is very weak. The authors need to focus on their findings and what is the implication of their study rather than mention findings from other publications. This in fact can be moved to the introduction.
The conclusion in the abstract and the conclusions in the main body of the paper are not consistent. The manuscript requires extensive revisions and has to be improved in clarity for it to have any significant meaning.
Author Response
Pané et al in their study titled ‘Bariatric surgery outcomes in patients with kidney transplantation’ carried out a retrospective study to compare metabolic profiles of patients who had received a kidney transplantation (KTx) either before or after BS. The idea notwithstanding the authors fail to impress the importance of carrying out this study and there are major concerns.
The introduction can be improved by adding points for the discussion that can serve as a background. The authors need to explain their results better and highlight the importance of their findings.
The write up from 107 to 115 is confusing. Figure 1, Table 1 and Table 2 could be represented in a better manner. In the present form they are very confusing.
The eGFR and Sr Creat has been mentioned for the post KTx group and not in the pre KTx group.
The discussion is very weak. The authors need to focus on their findings and what is the implication of their study rather than mention findings from other publications. This in fact can be moved to the introduction.
The conclusion in the abstract and the conclusions in the main body of the paper are not consistent. The manuscript requires extensive revisions and has to be improved in clarity for it to have any significant meaning.
Thank you for making us aware of the main drawbacks of our manuscript.
We apologize for the poor background and weak discussion/conclusions provided in the first version of our original manuscript. In order to solve these weaknesses, we have reformulated both the introduction and discussion, highlighting how our results can contribute to gain a possibly small, but useful insight into the controversies regarding the surgical management of obesity for kidney transplant candidates and recipients. In fact, multiple areas of uncertainty on this specific topic have been recently published by the DESCARTES Working Group of ERA.
We certainly agree with your concerns attending the phrasing of lines 107 to 115 and also, the format of both tables and figure 1. In order to transmit a clearer and more concise idea of our central results, we have reconsidered the presentation of the main tables and offered readers a precise reference to supplementary material. Considering figure 1, we believe that the explanation provided in the main text can help the authors to get its main message: weight regain appears to be a common pattern in the BS-post-KTx group.
In relation to your comment about serum creatinine, as all patients in the BS-pre-KTx group were on renal replacement therapy (haemodialysis), no renal function data is provided in Table 1. You can find this clarification at the bottom of the table: “Renal function data are not shown in BS-pre-KTx because all the patients were on hemodialysis”.
We are greatly thankful for your all suggestions and comments, which have completely improved the quality of the manuscript and made it much clearer.
Reviewer 3 Report
Unfortunately, in the bariatric field comparing a group of patients of 6 vs. 5 is inappropriate for assessing post-op outcomes because the complications are low in any subgroup, so this can be published as a series of cases but impossible to compare both groups and make a statement and conclusion without adequate STATISTICAL ANALYSIS due to low number of patients, FOR THAT REASON all the p Values are not significant
Author Response
Unfortunately, in the bariatric field comparing a group of patients of 6 vs. 5 is inappropriate for assessing post-op outcomes because the complications are low in any subgroup, so this can be published as a series of cases but impossible to compare both groups and make a statement and conclusion without adequate STATISTICAL ANALYSIS due to low number of patients, FOR THAT REASON all the p Values are not significant
We are grateful for your comment and we truly agree with your suggestion. As stated in the information in the “Material and Methods” section, because of the reduced group sizes, a descriptive approach was prioritized.
Following your advice, we have suppressed the p-values presented in table 2 since the number of events (complications, mortality, etc.) in every group is scarce and so, offering a “p-value” might confuse our readers when drawing firm conclusions. However, we have decided to maintain them in the descriptive tables (table 1 and supplementary material) just as a secondary approach to test groups comparability. In fact, we have lessened its visibility by reducing the letter size and columns width as we do not consider the “p-value” the leading message.
Round 2
Reviewer 2 Report
Although the authors have attempted to make the revisions, these were not as suggested. The idea of the study has potential, but the manuscript needs to be worked on seriously for it to be considered for publication.
